# Analysis of the Influencing Factors of Drought Events Based on GRACE Data under Different Climatic Conditions: A Case Study in Mainland China

**Lilu Cui** [1,2]📧, **Cheng Zhang** [1], **Chaolong Yao** [3,*], **Zhicai Luo** [4], **Xiaolong Wang** [5]📧 **and Qiong Li** [6]

1 School of Architecture and Civil Engineering, Chengdu University, Chengdu 610106, China; cuililu@cdu.edu.cn (L.C.); lilucui@whu.edu.cn (C.Z.)
2 School of Geodesy and Geomatics, Wuhan University, Wuhan 430079, China
3 College of Natural Resources and Environment, South China Agricultural University, Guangzhou 510642, China
4 MOE Key Laboratory of Fundamental Physical Quantities Measurement & Hubei Key Laboratory of Gravitation and Quantum Physics, PGMF and School of Physics, Huazhong University of Science and Technology, Wuhan 430074, China; zcluo@hust.edu.cn
5 Nanning Exploration & Survey Geoinformation Institute, Nanning 530022, China; wwxyzwxl@whu.edu.cn
6 School of Civil Engineering and Geomatics, Southwest Petroleum University, Chengdu 610500, China; qiongli@swpu.edu.cn
* Correspondence: clyao@scau.edu.cn; Tel.: +86-136-4075-1027

**Abstract:** The occurrence of droughts has become more frequent, and their intensity has increased in mainland China. With the aim of better understanding the influence of climate background on drought events in this region, we analyzed the role of the drought-related factors and extreme climate in the formation of droughts by investigating the relationship between the drought severity index (denoted as GRACE-DSI) based on the terrestrial water storage changes (TWSCs) derived from Gravity Recovery and Climate Experiment (GRACE) time-variable gravity fields and drought-related factors/extreme climate. The results show that GRACE-DSI was consistent with the self-calibrating Palmer Drought Severity Index in mainland China, especially for the subtropical monsoon climate, with a correlation of 0.72. Precipitation (PPT) and evapotranspiration (ET) are the main factors causing drought events. However, they play different roles under different climate settings. The regions under temperate monsoon climate and subtropical monsoon climate were more impacted by PPT, while ET played a leading role in the regions under temperate continental climate and plateau mountain climate. Moreover, El Niño–Southern Oscillation (ENSO) and North Atlantic Oscillation (NAO) events mainly caused abnormalities in PPT and ET by affecting the strength of monsoons (East Asian and Indian monsoon) and regional highs (Subtropical High, Siberian High, Central Asian High, etc.). As a result, the various affected regions were prone to droughts during ENSO or NAO events, which disturbed the normal operation of atmospheric circulation in different ways. The results of this study are valuable in the efforts to understand the formation mechanism of drought events in mainland China.

**Keywords:** GRACE; drought; mainland China; extreme climate; climatic conditions

## 1. Introduction

Drought is a severe natural hazard event on a global scale characterized by terrestrial water deficit. It has a negative impact on socioeconomic development, crop failure, ecosystems, and the lives of people [1–3]. Therefore, research on the influencing factors of drought events are of great significance for establishing early warning, strengthening water resources management and reducing disaster losses [4]. The drought index has always been used as a quantitative indicator to characterize the drought events due to its simple and easy-to-understand characteristics [5]. At present, the commonly used drought indices

are mainly the palmer drought severity index (PDSI) [6], the standardized precipitation index (SPI) [7] and the standardized runoff index (SRI) [8]. These traditional drought indices are mainly calculated based on the long-term accumulation of drought-related data such as precipitation (PPT), evapotranspiration (ET), temperature, etc. The drought-related data is provided by the meteorological stations. However, an insufficient number and uneven distribution of these stations in some regions leads to the inability to obtain surface data with high spatial resolution [9]. Furthermore, the traditional technical approaches not only require a great deal of construction and maintenance, and they can only observe parts of the hydrological component in the terrestrial water cycle [10]. Therefore, it is impossible to explain the cause of drought from the perspective of the entire terrestrial water cycle. These problems also appear in the traditional drought indices.

Since 2002, the Gravity Recovery and Climate Experiment (GRACE) mission [11] has provided monthly data on Earth's gravity field to infer the total terrestrial water storage change (TWSC) including surface and subsurface hydrological components [12]. Many research works have proved that the GRACE data can detect regional drought events and assess drought-associated losses, in regions such as the Amazon River [10,13,14], Yangtze River [9,15–17], Southeastern China [18–20] and State of Texas [21]. Therefore, some scholars have used GRACE TWSC data to construct drought indexes to achieve more accurate detection and assessment of local drought. Yirdaw et al. [22] derived the total storage deficit index (TSDI) to characterize the drought events in the Katchewan River during 2002 and 2003. Wang et al. [23] used the GRACE TWS anomaly index (TWSI), PPT anomaly index and vegetation anomaly index to detect drought events in the Haihe River basin from January 2003 to January 2013. The results indicate that TWSI is more suitable than traditional indices to monitor these drought events. Yi et al. [24] constructed the GRACE-based hydrological drought index (GHDI) to monitor the drought events in the United States from 2003 to 2012. The results indicate that the GHDI has a good correlation with PDSI over the United States. Sinha et al. [25] used the water storage deficit index (WSDI) to assess drought events in India. The results illustrate the validity and reliability of WSDI in quantifying the characteristics of large-scale drought events. Zhao et al. [26] developed the GRACE-based drought severity index (GRACE-DSI) to capture the major drought events worldwide, and GRACE-DSI showed good temporal and spatial agreement with PDSI and the standardized precipitation evapotranspiration index (SPEI). The above studies show that the GRACE-based drought index is a valuable tool for the detection and assessment of hydrological drought.

When people conduct in-depth research on drought-related data, they have a certain understanding of the driving mechanisms of drought. Li et al. [19] and Wu et al. [27] indicate that the extreme drought in Southwest China in 2010 was mainly caused by insufficient PPT, and that excessive ET played a secondary role. Panisset et al. [28] explain that anomalous PPT deficit and solar radiation anomalies were the main factors leading to the three drought events in the Amazon basin in 2005, 2010 and 2015. Zhang et al. [17] studied the two drought events that occurred in the Yangtze River basin in 2006 and 2011. The results showed that there was a certain connection with droughts and El Niño–Southern Oscillation (ENSO) in this region, and the TWSC in the lower reaches was more sensitive to the change in ENSO than the TWSC in the upper and middle reaches. However, the above studies mainly focused on a certain local drought event, and did not consider the influence of the regional climate background on the drought event.

Mainland China (MC) is the region with the most frequent drought disasters worldwide; these disasters have brought huge losses to the region, and local drought disasters occur almost every year in the region [29]. Therefore, we used MC as a research region to study the drought events that occurred from April 2002 to June 2017 in four different climate regions based on GRACE-DSI data. We calculated the correlation coefficients between PPT, runoff, ET soil water storage and GRACE-DSI in the four different climate regions to analyze the impact of the different climate backgrounds on drought events. We also carried out a statistical analysis of the PPT, ET and GRACE-DSI during ENSO and

North Atlantic Oscillation (NAO) events to discuss the influence of extreme climate on drought events in different climate regions. These research results can help to understand the driving mechanisms of drought events and provide early warning of drought disaster.

## 2. Study Area

MC is located approximately within 19° N–53° N and 73° E–135° E, and has an area of about 9.6 million km$^2$. Its digital elevation model shows that the terrain is like a ladder, gradually descending from west to east (Figure 1).

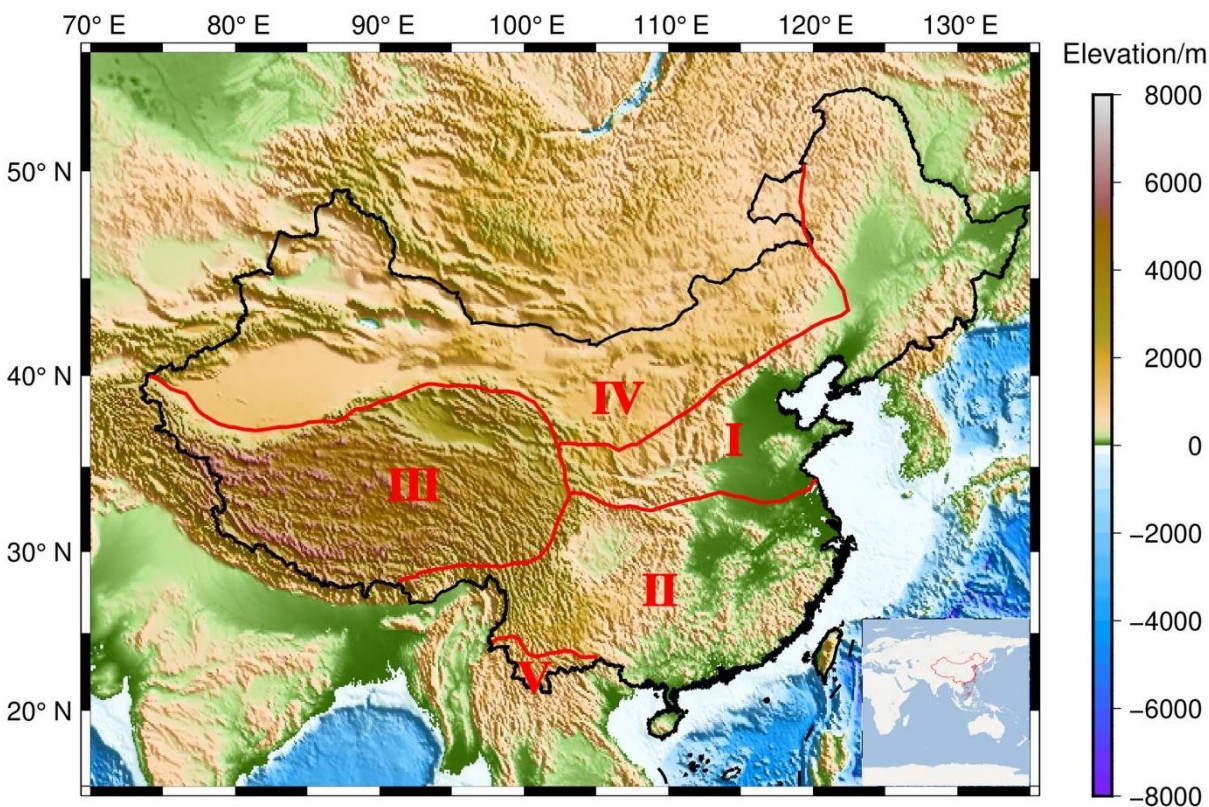

**Figure 1.** Digital elevation model of mainland China. The different climate regions are marked on the map: temperate monsoon climate (I), subtropical monsoon climate (II), plateau mountain climate (III), temperate continental climate (IV) and tropical monsoon climate (V).

China is a vast territory with a wide span of latitudes and many areas that are far from the sea. In addition, the terrain types and mountain directions are diverse, resulting in a diversity of temperature, PPT and climates formed. The Eastern region has a monsoon climate, the Northwest region has a temperate continental climate and the Qinghai-Tibet Plateau has a plateau mountain climate. There are also humid regions, semi-humid regions, semi-arid regions and arid regions. Furthermore, it is one of the countries with the most rivers in the world. There are more than 1500 rivers with an area larger than 1000 km$^2$ in MC, including the Yangtze River, Yellow River, Pearl River, Huaihe River, Liaohe River and Songhua River. Most rivers are located in Eastern and Southern China, and a few rivers are located in Northwest China. The ET gradually decreases from Southeast to Northwest China. The distribution of precipitation decreases sharply from the Southeastern (>3000 mm) to the Northwestern region (<50 mm) [30]. The occurrence of droughts and floods has become more frequent, and the intensity of drought and floods has also increased [31]. According to the climate type, MC can be divided into five parts (Figure 1), namely temperate monsoon climate (I), subtropical monsoon climate (II), plateau mountain climate (III), temperate continental climate (IV) and tropical monsoon climate (V). As the

area of region V is relatively small, it is not suitable for GRACE detection; we focus on regions I–IV in this study.

## 3. Data and Methods

### 3.1. GRACE TWSC

The GRACE RL06 monthly spherical harmonic (SH) coefficients product (truncated to degree and order 60) was provided by the Center for Space Research at the University of Texas at Austin, and was used to calculate TWSCs in MC for the period from April 2002 to December 2016.

The GRACE data were preprocessed as follows: C20 coefficients were replaced with those derived by satellite laser ranging [32]. Degree-1 coefficients were replaced using Swenson's results [33]. Filter processing combining a 300 km fan filter [34] and a PM36 de-correlation filter [35] was performed to weaken high-frequency and correlated errors. Due to the influence of order truncation and filter processing, there are leakage errors in the hydrological signals derived from GRACE data. The single scale factor method was used to calibrate GRACE-based TWSC results to restore loss signals [36].

### 3.2. GLDAS Model

The GLDAS is a global high-resolution land surface model published jointly by the Goddard Space Flight Center at NASA and the National Centers for Environmental Prediction at the National Oceanic and Atmospheric Administration (NOAA). It incorporates space- and ground-based observation and uses data assimilation techniques [37]. The monthly soil moisture (SM) and runoff data with a $1° \times 1°$ spatial resolution were provided by three GLDAS-2.1 models (Noah, the variable infiltration capacity model and the catchment land surface model). The SM and runoff data were the average of these three models' data, and the time span was from April 2002 to December 2016.

### 3.3. In Situ Precipitation (PPT) Data

Monthly gridded precipitation data for the time period between April 2002 and December 2016 provided by the China National Meteorological Science Data Center and sorted by the National Meteorological Information Center with spatial resolution $0.5° \times 0.5°$ were used for the analysis.

### 3.4. ET Data

ET is estimated according to a water balance equation [38,39]. Its expression is as follows:

$$\text{ET} = \text{P} - \text{R} - \text{TWSC} \tag{1}$$

where P is PPT, R is the averaged runoff derived from GLDAS-2.1, and TWSC is derived from GRACE data.

### 3.5. Self-Calibrating Palmer Drought Severity Index (SCPDSI) Data

The SCPDSI data [40], a meteorological drought index, can evaluate the water loss caused by the imbalance of surface water supply and demand [41,42], and is provided by the Climate Research Unit at University of East Anglia. In this study, we extracted the relevant gridded data from April 2002 to December 2016 with spatial resolution $0.5° \times 0.5°$ in MC. The severity of drought events can be classified as shown in Table 1 [43].

**Table 1.** The grades of SCPDSI drought classification.

| Type | SCPDSI | Type | SCPDSI |
|---|---|---|---|
| Extreme Drought | $\leq -4.0$ | Light Drought | $-1.0 \sim -2.0$ |
| Heavy Drought | $-3.0 \sim -4.0$ | No Drought | $\geq -1.0$ |
| Moderate Drought | $-2.0 \sim -3.0$ | | |

### 3.6. Extreme Weather Index Data

The ENSO is an abnormal phenomenon characterized by ocean surface warming or higher sea surface temperatures occurring in the Equatorial Eastern and Middle Pacific, and is able to influence the global atmospheric circulation, causing abnormal temperature and PPT [44,45]. The monthly Niño 3.4 index data indicate the magnitude of ENSO, which is provided by the NOAA. An El Niño event is designated as the occurrence of an ENSO index greater than or equal to 0.5 for 5 consecutive months, while a La Niña event is defined as an ENSO index less than or equal to −0.5 for 5 consecutive months [44,46].

The NAO is a kind of atmospheric circulation change that occurs in the middle and high latitudes of the Northern Hemisphere in winter, which reflects the atmospheric mass changes between Iceland Depression and Azores High in the North Atlantic. The NAO index can reflect the changes in the Iceland Depression and Azores High [47], and is also provided by the NOAA.

### 3.7. Calculation of GRACE-DSI Data

Based on the gridded GRACE TWSCAs estimated in Section 3.1, GRACE-DSI is the standardized GRACE-based TWSC as follows [26]:

$$\text{GRACE-DSI}_{i,j} = \frac{\text{TWSC}_{i,j} - \text{TWSC}_j^{mean}}{\sigma_j} \tag{2}$$

where $\text{TWSC}_{i,j}$ is TWSC in the $i$th year and $j$th month. $i$ is a specific year from 2002 to 2016, and $j$ is a specific month from January to December. $\text{TWSC}_j^{mean}$ and $\sigma_j$ are the average and standard deviation of the TWSC in month $j$, respectively. This index can be used to detect drought and abnormally wet events. According to the size of GRACE-DSI values, drought events can be classified as shown in Table 2 [26]. Due to the truncation degree and filtering effect, the spatial resolution of GRACE-DSI grid data is 350 km [48].

**Table 2.** GRACE-DSI drought grades classification.

| Type | GRACE-DSI | Type | GRACE-DSI |
|---|---|---|---|
| Exceptional Drought | $\leq -2.0$ | Moderate Drought | $-1.3\sim-0.8$ |
| Extreme Drought | $-2.0\sim-1.6$ | Light Drought | $-0.8\sim-0.5$ |
| Severe Drought | $-1.6\sim-1.3$ | No Drought | $\geq -0.5$ |

### 3.8. The Extraction of Anomaly Signal

To discuss the relationship between GRACE-DSI and hydrological components, it is necessary to extract the anomaly signal from the original signal of each hydrological component. The time series of the original data can be decomposed into a long-term trend change term, seasonal term and anomaly term. The expression is as follows [36]:

$$Data(t) = a_0 + a_1 t + a_2 \cos(2\pi t) + a_3 \sin(2\pi t) + a_4 \cos(4\pi t) + a_5 \sin(4\pi t) + \varepsilon \tag{3}$$

where $Data(t)$ is the original data; $t$ is the time; $\varepsilon$ is the residual signal; and $a_0, a_1, a_2, a_3, a_4, a_5$ are the pending parameters. $a_0$ is a constant; $a_1$ is the long-term trend change term; and $a_2, a_3, a_4, a_5$ are the seasonal signals. Therefore, the expression of the anomaly signals is:

$$Data_{anomaly}(t) = Data(t) - [a_1 t + a_2 \cos(2\pi t) + a_3 \sin(2\pi t) + a_4 \cos(4\pi t) + a_5 \sin(4\pi t)] \tag{4}$$

## 4. Results and Analysis

### 4.1. Comparison of GRACE-DSI and SCPDSI

To verify the drought detection ability of GRACE-DSI, we compared the temporal and spatial distribution of GRACE-DSI and SCPDSI (Figures 2 and 3). From Figure 2, it can be seen that the two drought indices show good consistency in general, with a correlation of 0.66 (Table 3). There were four episodes with long-term negative GRACE-DSI. The

first time period was from August 2002 to March 2003, with the minimum value (−0.64) occurring on March 2003. Although SCPDSI showed similar changes during this period, the negative value (−0.07) was small and only appeared for one month (October 2002). The values of GRACE-DSI were mostly negative during the second period from June 2006 to February 2010, coinciding with SCPDSI. The minimum values of these indices appeared on April 2008 (GRACE-DSI, −0.62) and August 2006 (SCPDSI, −0.71), respectively. During the third period from March 2011 to April 2012, the minimum value of GRACE-DSI (−0.40) was smaller than that of SCPDSI (−0.31), which appeared on January 2012 and July 2011, respectively. For the fourth dry period from September 2013 to July 2015, the minimum value (−0.43) of GRACE-DSI occurred in January 2015 and the one of SCPDSI was −0.35 in April 2014.

Figure 3 shows the spatial distribution of GRACE-DSI and SCPDSI from October 2009 to September 2010. It can be seen in Figure 3 that the two drought indices had similar spatial distribution. In October 2009, most parts of MC were in the dry state, and only parts of Qinghai Province and the northern part of Northeast China were in the humid state. From November 2009 to January 2010, the arid regions were gradually decreasing, and the degree of dryness was also decreasing. January 2010 was the period with the fewest arid regions. By February 2010, the arid regions suddenly expanded, which was mainly concentrated in Southwest China, eastern Xinjiang, southern Northeast China and northern North China. The humid regions were concentrated in Qinghai Province, Northeast China and the Southeast Coastal region. Subsequently, the arid regions gradually decreased and the humid regions gradually increased. This trend continued until September 2010. The temporal and spatial changes of GRACE-DSI and SCPDSI in Southwest China are consistent with the severe drought event that occurred in the same period in the region [19].

From the perspective of different climate types, the highest correlation (0.72) between GRACE-DSI and SCPDSI was found in region II under a subtropical monsoon climate (see Table 3). In this region, the precipitation accounted for a relatively large proportion of the entire water cycle. The lowest correlation (0.29) was found in region III under the plateau mountain climate, implying a complex mechanism of droughts in this region. Since vegetation, snow and other terrestrial surface hydrological components are not explicitly processed in the SCPDSI [40], the correlations (0.42) between the two drought indices were the same in the other three regions. By comparing the temporal and spatial distribution of two drought indices and their correlation coefficients (Table 3, Figures 2 and 3), it can be seen that GRACE-DSI could detect the drought events.

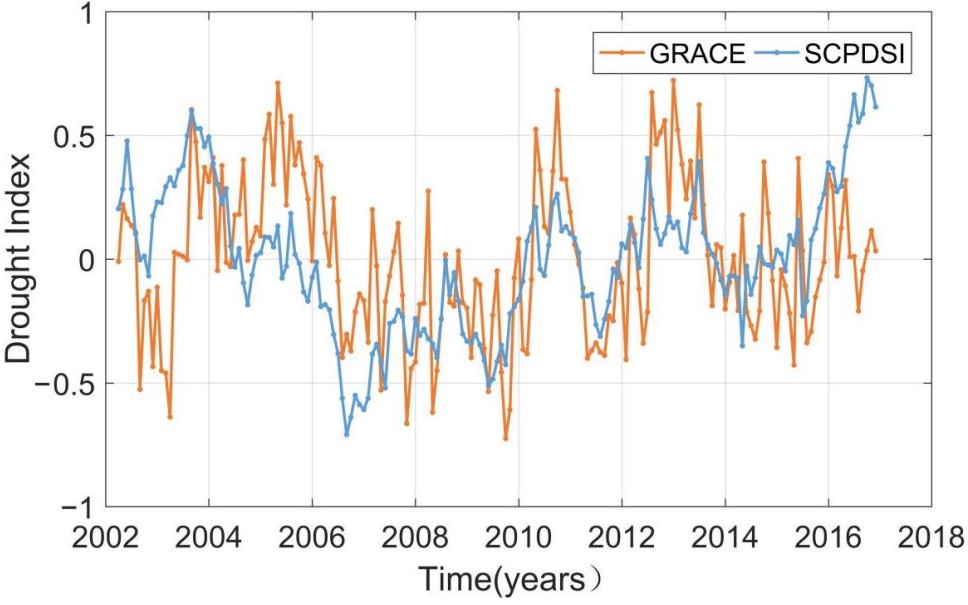

**Figure 2.** Time series of GRACE-DSI and SCPDSI.

**Table 3.** The correlation coefficients between GRACE-DSI and SCPDSI.

| Area | I | II | III | IV | V | MC |
|---|---|---|---|---|---|---|
| Correlation Coefficient | 0.42 | 0.72 | 0.29 | 0.42 | 0.42 | 0.66 |

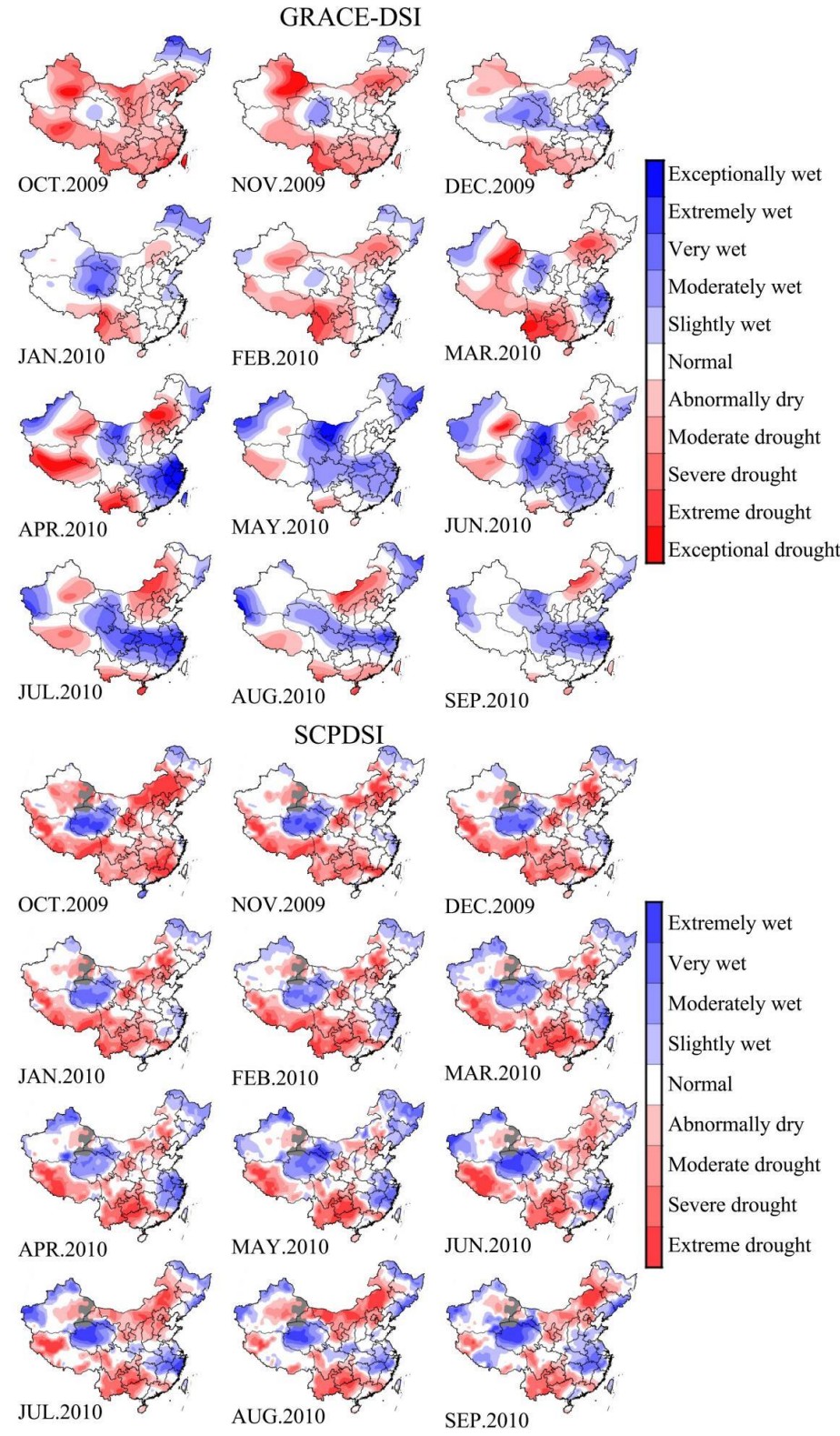

**Figure 3.** Spatial distribution of GRACE-DSI and SCPDSI concerning the October 2009–September 2010 period.

*4.2. The Analysis of Factors Affecting GRACE-DSI*

To study the impact of SM, PPT, ET and runoff on drought events, we calculated the correlation coefficients between GRACE-DSI/SCPDSI and SM, PPT, ET and runoff anomaly. The anomaly signals were extracted according to Equation (4). For a more intuitive comparison and analysis, the following will elaborate on different climate regions. As the tropical monsoon climate region is very small, it is not conducive to the detection of GRACE satellites; therefore, it is considered here.

4.2.1. Temperate Monsoon Climate (Region I)

Figure 4 shows the time series of GRACE-DSI and SM, PPT, ET and runoff anomaly. The SM and runoff had a relatively consistent change trend with GRACE-DSI and the consistency of SM with GRACE-DSI was higher. However, the time series of GRACE-DSI and SM had opposite change trends. This is also supported by the correlation coefficient results (Table 4). The above results indicate that the SM is the most important factor affecting the GRACE-DSI. This is consistent with the definition of hydrological drought—that is, an imbalance between the supply and demand of soil water storage causes a drought event [49].

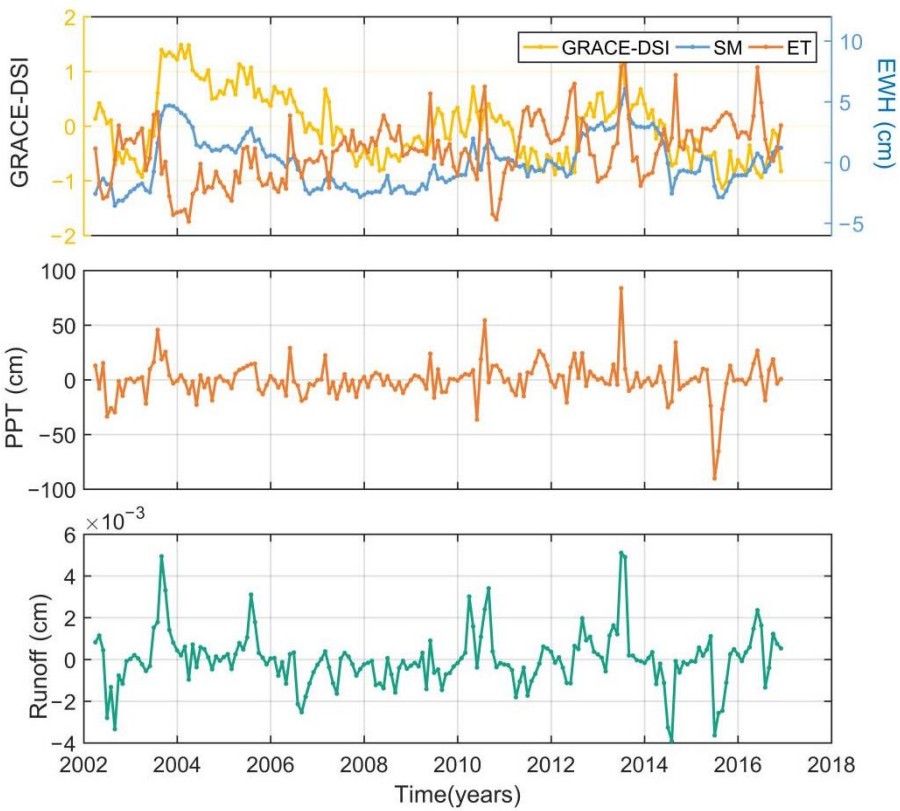

**Figure 4.** The time series of GRACE-DSI, PPT, SM, runoff and ET anomaly under the temperate monsoon climate.

**Table 4.** The correlation coefficients between GRACE-DSI and hydrological components.

| Hydrological Component | PPT | SM | Runoff | ET |
|:---:|:---:|:---:|:---:|:---:|
| GRACE-DSI | 0.18 | 0.75 | 0.37 | −0.61 |

To analyze the interaction between the terrestrial hydrological components under the temperate monsoon climate, the correlation coefficients between SM, PPT, ET and runoff were calculated (Table 5). Table 5 shows that there was a strong correlation between PPT

and runoff, and between runoff and SM, implying an obvious water transport channel between PPT, SM and runoff—that is, PPT affects runoff, runoff affects SM. Considering this with Table 4, we can see that the connection between PPT and GRACE-DSI is not strong, but PPT exerts the effect on GRACE-DSI through runoff and SM. Considering this alongside the strong correlation between ET and GRACE-DSI, it explains that PPT and ET are the main factors affecting the occurrence of drought under this climate. Since the correlation coefficient between SM and GRACE-DSI was larger than that between ET and GRACE-DSI, it can be said that PPT has a larger impact on drought than ET. In a word, PPT is the mainstay and ET is the supplement during the formation of drought events.

**Table 5.** The correlation coefficients between different hydrological components.

| Hydrological Component | PPT | SM | Runoff | ET |
|---|---|---|---|---|
| PPT | 1 | 0.29 | 0.61 | 0.32 |
| SM | 0.29 | 1 | 0.54 | −0.31 |
| Runoff | 0.61 | 0.54 | 1 | 0.13 |
| ET | 0.32 | −0.31 | 0.13 | 1 |

Usually, PPT and ET are vulnerable to extreme weather, so it is necessary to consider the impact of extreme weather on drought events. Figures 5 and 6 show the performance of the time series of GRACE-DSI, PPT and ET during the ENSO and NAO events, respectively. From Figure 6, a total of five El Niño events and five La Niña events occurred during the study period. The five El Niño events caused abnormal decreases in PPT, while three El Niño events also led to abnormal increases in ET. Previous studies [50,51] indicate that because of the abnormal decrease of sea surface temperature in the Western Pacific (El Niño event), the East Asian summer monsoon weakened, causing the Western Pacific Subtropical High and rain belt to move southward, resulting in less PPT and higher temperatures in Northern China. The results in this paper provide scientific support for these results. Due to less PPT and more ET, severe drought events occurring during the El Niño events from June 2002 to February 2003 and from April 2015 to April 2016. While the two El Niño events from July 2004 to January 2005 and from September 2006 to January 2007 caused an abnormal decrease in PPT, the ET also showed an abnormal decrease. The drought did not occur under the mutual offset of PPT and ET. The El Niño events from July 2009 to March 2010 led to less PPT and more ET, but perhaps because of the degree of PPT reduction and the minor increase in ET, the combined effect was not enough to cause a drought. According to the above analysis, we found that an El Niño event can indeed cause an abnormally low PPT and an increase in ET, but its impact on PPT is slightly greater than that on ET. Whether an El Niño event will cause drought is the result of its combined effect on PPT and ET. The intensity of this effect needs to reach a certain level in order to induce drought.

Three La Niña events led to an abnormal increase in PPT, and two La Niña events caused an abnormal reduction in ET. Ma [52] indicates that when a La Niña event occurs, the effects are simply the opposite of an El Niño event. At the time, the Western Pacific Subtropical High and rain belt moved northward with the strengthening of the East Asian Monsoon. Northern China showed higher PPT and higher temperature. It can be seen from Figure 6 that the La Niña event mainly affected PPT. Drought events occurred during the La Niña events from July 2007 to June 2008 and from August 2016 to December 2016. According to the previous studies, the cause of drought from 2007 to 2008 was that the northern part of China was in an interdecadal climate with high temperature and low PPT at that time, which caused a high probability of drought in Northeast China and North China [53]. The other drought event was the result of the interaction between the abnormal high pressure in Baikal Lake and Central Siberia and the abnormal low pressures in North China [54]. Due to the interaction of the above-mentioned high and low pressure, the central and eastern regions of China were controlled by the extremely strong dry and cold air flow. Under the control of the air flow, the water vapor transport was reduced, which in

turn caused an abnormal decrease in PPT. This explains that the two drought events above had little to do with the La Niña events.

Under the temperate monsoon climate, the ENSO cycle mainly affects the location of the Western Pacific Subtropical High through the strength of the East Asian Monsoon. The location of Western Pacific Subtropical High determines the amount of PPT and the temperature. The temperature affects the amount of ET. Less PPT and more ET will increase the probability of drought during an El Niño event. The situation is the opposite during a La Niña event.

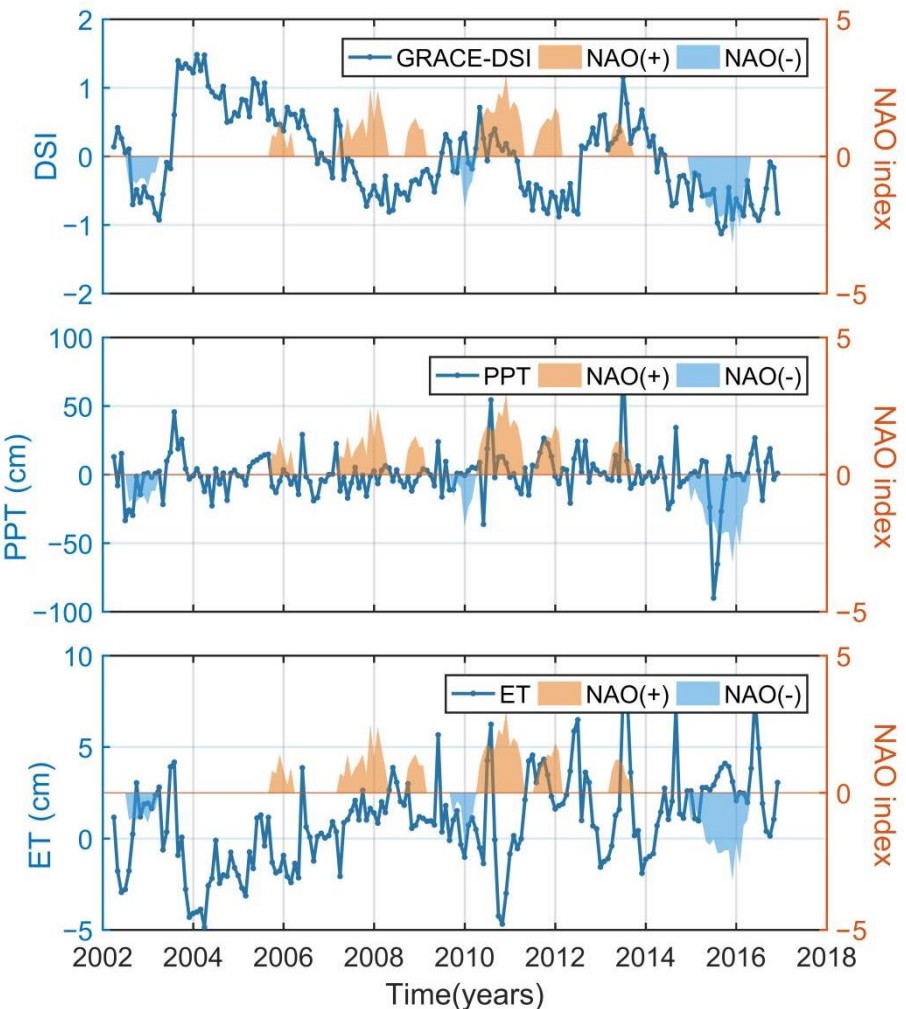

**Figure 5.** GRACE-DSI, PPT anomaly and ET anomaly during NAO events.

The relationship between NAO and GRACE-DSI, PPT and ET were analyzed in this paper (see Figure 5). There were three negative NAO events and seven positive NAO events during the study period. We found that the drought events occurred during the negative NAO events, and they were all caused by less PPT and more ET. Wu et al. [55] indicate that the NAO index has an inverse relationship with the range of Siberian High. When the NAO index was abnormally low, the Siberian High enhanced and its impact scope expanded, which cause the rain belt to move southward and there was less PPT and more ET in Northern China. This coincides with the results in this paper.

When the NAO index was abnormally high, the situation was the opposite. However, from Figure 5, it can be observed that there were three drought events during the positive NAO events. Among them, the drought event from August 2007 to August 2008 was caused by the background of climate, as explained in the previous section, and the one from October 2008 to February 2009 was affected by an El Niño event, indicating that the impact of the El Niño event may have exceeded the positive NAO event. However, the

specific formation mechanism of the above drought event is relatively complicated, and conclusions cannot be drawn regarding this as of yet. The drought event from August 2011 to February 2012 was mainly caused by abnormal atmospheric circulation. At the same time, the Western Pacific Subtropical High was located to the south, which was not conducive to the transportation of water vapor. In addition, there was a strong sinking movement and low humidity in this region, which is conducive to the development and continuation of drought [56].

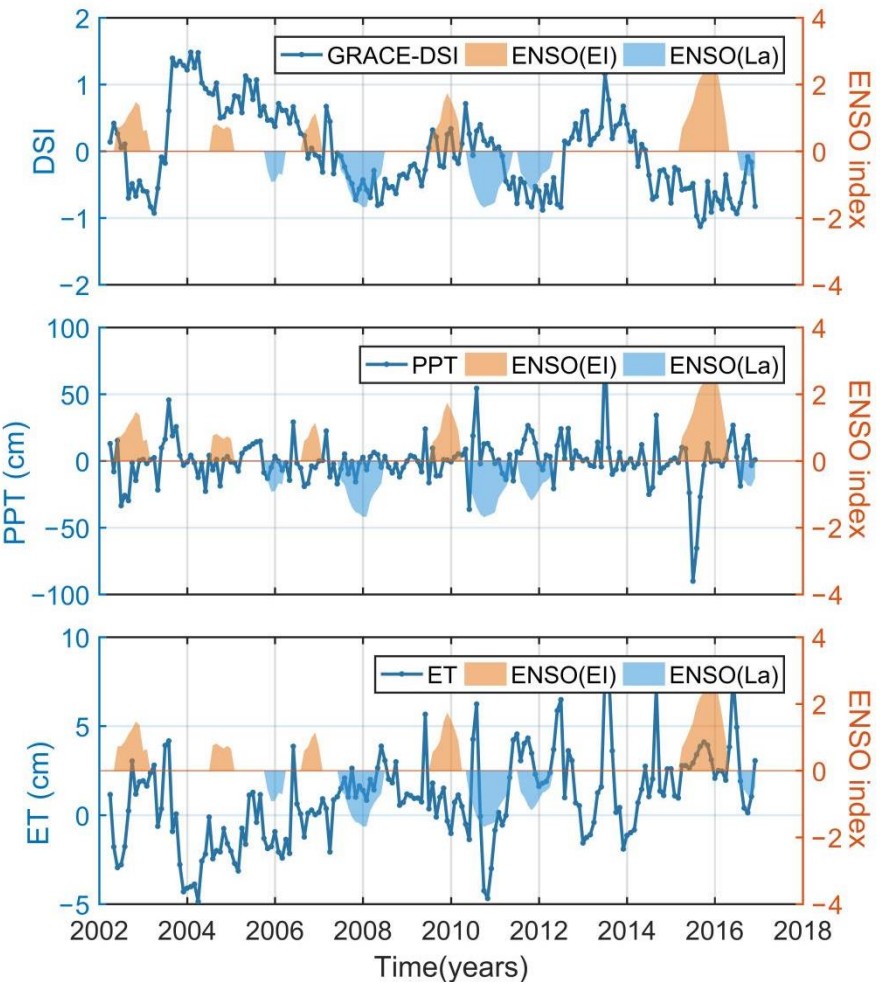

**Figure 6.** GRACE-DSI, PPT anomaly and ET anomaly during ENSO events (El, El Niño, orange; La, La Niña, blue).

Ineson et al. and Graf et al. [57,58] indicate that ENSO signal is propagated to the stratosphere by upward movement, and then transmitted to the North Atlantic region through the "subtropical bridge" mechanism in the stratosphere, which causes the NAO response. El Niño is a negative NAO event, while La Niña is a positive NAO event. This is consistent with the significant negative correlation (−0.88) between ENSO and NAO indices in this paper. Comparing Figures 5 and 6, we can see that there were seven ENSO events accompanied by NAO events. Among the seven events mentioned above, there were three El Niño events, and the negative NAO events occurred at the same time; additionally, four La Niña events and positive NAO events occurred together. The above results provide strong data support for Chen et al. [59]. However, there were three ENSO events that did not cause corresponding NAO events. This may be because these three ENSO events were not Central Pacific (CP) events. According to the study results of Zhang et al. [60], there is a significant relationship between CP ENSO events and NAO events.

### 4.2.2. Subtropical Monsoon Climate (II)

Figure 7 shows the time series of GRACE-DSI, PPT, SM, runoff and ET anomaly under the subtropical monsoon climate. We found that GRACE-DSI, SM and runoff had similar change trends, which was also confirmed by the correlation coefficient results (Table 6). Unlike the results under the temperate monsoon climate, the correlation coefficients indicate that there was no significant correlation between GRACE-DSI and ET, but GRACE-DSI had a close connection with runoff. This may be related to the sufficient PPT and numerous rivers in this region. We calculated the correlation coefficients between the four hydrological components (Table 7). It can be seen that the way in which PPT affected SM was the same as in the temperate monsoon climate, and the impacts of PPT on runoff, and of runoff on SM, were much greater than was observed in the temperate monsoon climate. This indicates that PPT plays a leading role in drought events, and the impact of ET is small under the subtropical monsoon climate.

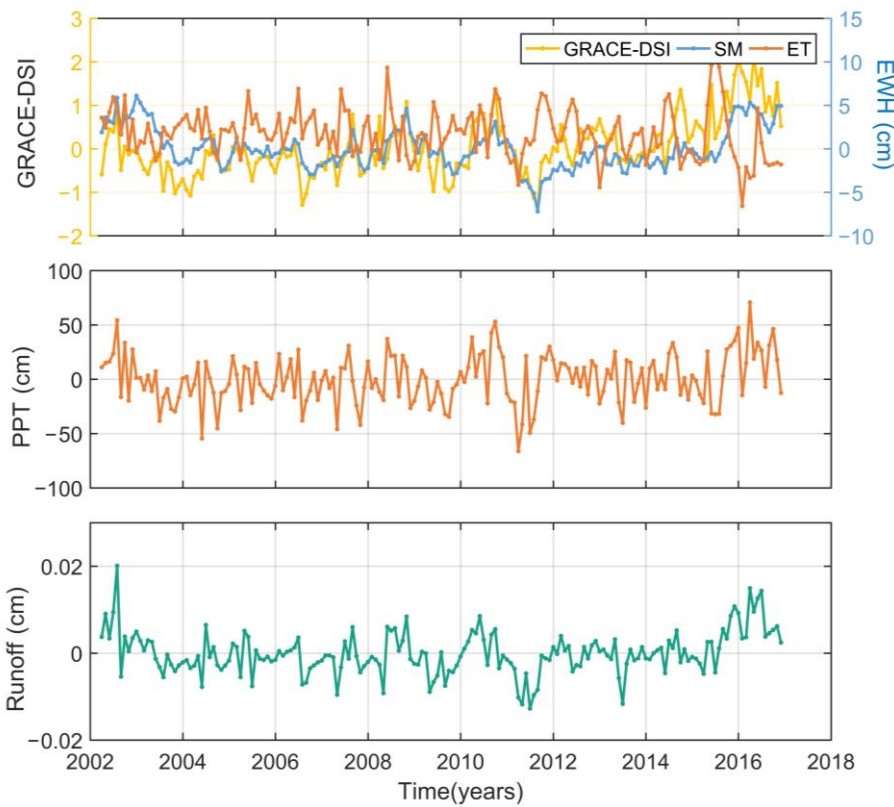

**Figure 7.** The time series of GRACE-DSI, PPT anomaly, SM anomaly, runoff anomaly and ET anomaly under the subtropical monsoon climate.

**Table 6.** The correlation coefficients between GRACE-DSI and hydrological components.

| Hydrological Component | PPT | SM | Runoff | ET |
|:---:|:---:|:---:|:---:|:---:|
| GRACE-DSI | 0.47 | 0.65 | 0.62 | −0.24 |

**Table 7.** The correlation coefficients between different hydrological components.

| Hydrological Component | PPT | SM | Runoff | ET |
|:---:|:---:|:---:|:---:|:---:|
| PPT | 1 | 0.40 | 0.72 | 0.29 |
| SM | 0.40 | 1 | 0.73 | −0.23 |
| Runoff | 0.72 | 0.73 | 1 | 0.11 |
| ET | 0.29 | −0.23 | 0.11 | 1 |

Similarly, to discuss the influence of extreme climate on drought events and PPT, the change condition of GRACE-DSI and PPT anomaly time series during ENSO and NAO events are shown in Figures 8 and 9. Figure 8 shows that there were drought events during three La Niña events. These drought events were caused by low PPT. Previous studies have shown that the PPT in Southeastern China was greater than normal due to the southward shift of the PPT belt during La Niña events, while the situation was the opposite during the El Niño events [50,61]. There was no reduction in PPT during the other two La Niña events, and so drought events did not appear. This can be attributed to the large amount of PPT brought by typhoons [62]. However, drought events occurred during three El Niño events, and these three drought events occurred from July 2004 to January 2005, from September 2006 to January 2007 and from July 2009 to May 2010. The first drought event was mainly due to a lack of PPT caused by a lack of tropical cyclones [63]. The second drought event was mainly caused by the control of the Subtropical High in Southern China, and the increase and continuation of the Subtropical High in 2006 were closely related to the strengthening of atmospheric convection in the South China Sea and the abnormal heating field in the Bengal Bay [64]. The main reason for the drought from 2009 to 2010 was abnormal circulation. The Western Pacific Subtropical High was stronger than usual, and as a result the Indian Ocean water vapor was not transported to Southwest China. Therefore, there was less PPT in the region [65].

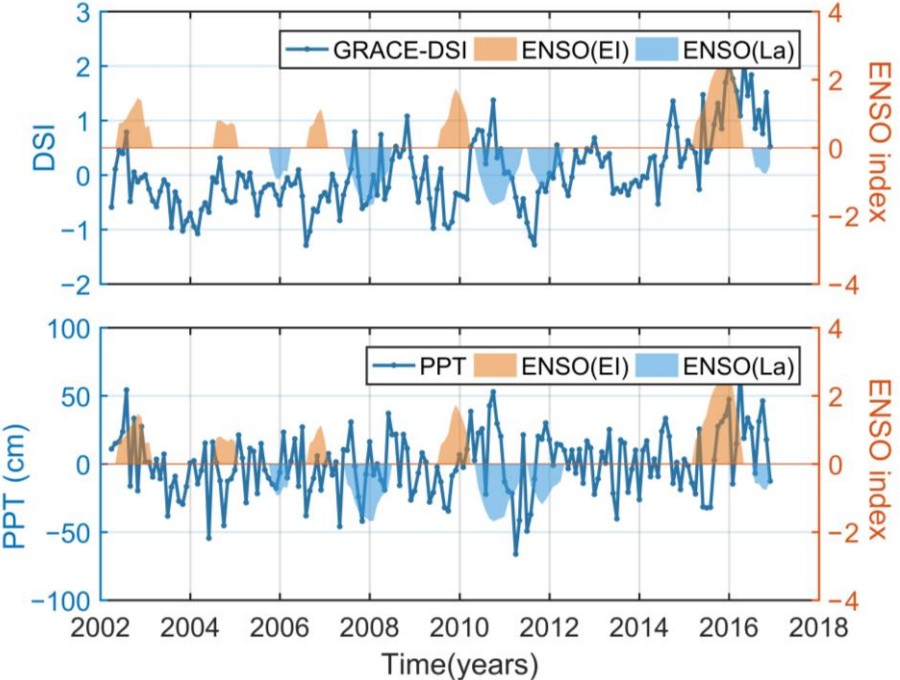

**Figure 8.** GRACE-DSI and PPT anomaly during ENSO events (El, El Niño; La, La Niña).

From Figure 9, it can be observed that the drought events occurred during four positive NAO events, which were caused by low PPT. A previous study indicates that a positive NAO event caused the rain belt to move north, so there was less PPT in this region [55]. The positive NAO event from October 2008 to February 2009 caused a reduction in PPT, but the peak and average values of GRACE-DSI were greater than 0 because of two large-scale waves in the winter 2008, which made the temperature lower than usual [66]. Therefore, severe drought did not appear under the interaction of reduced PPT and low temperature. There was only one drought during a negative NAO event from October 2009 to March 2010, which was caused by the Western Pacific Subtropical High. The specific reason is explained in the previous paragraph.

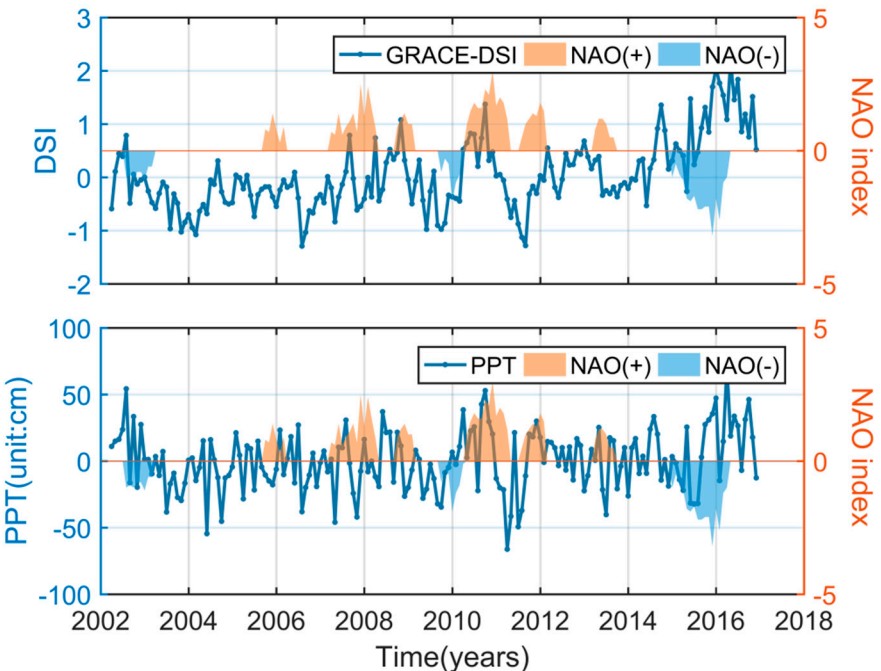

**Figure 9.** GRACE-DSI and PPT anomaly during NAO events.

4.2.3. Plateau Mountain Climate (III)

The time series of GRACE-DSI, PPT, SM, ET and runoff anomaly under the plateau mountain climate are shown in Figure 10. Only the ET and GRACE-DSI had a significant correlation, and it was negative. This is also confirmed by the results in Table 8. The change trends of runoff and PPT anomaly were relatively smooth. Table 9 shows that the correlations between the four hydrological components (SM, PPT, SM and runoff) were not strong, indicating that ET is the main factor causing drought events under the plateau mountain climate.

Figures 11 and 12 show the time series of GRACE-DSI and ET anomaly during ENSO and NAO events. Figure 11 shows that drought events occurred during all five El Niño events. There were four drought events caused by higher ET. Xu et al. indicated that ENSO events affect the surface temperature of the Tibetan Plateau by adjusting the strength of the Indian Ocean Monsoon [67]. In an El Niño year, the Indian Ocean Monsoon weakens and the surface temperature rises, leading to an increase in the possibility of drought events. In La Niña years, the opposite is true. However, drought events occurred during two La Niña events. The first drought event from July 2007 to June 2008 was due to a stronger Middle East Subtropical High. The warm and humid airflow from Bengal Bay could not reach the Tibetan Plateau under the control of this high [68]. The second drought from August to December 2016 was caused by the southward movement of an abnormal continental warm high and the northward movement of the Western Pacific Subtropical High. Under the influence of the above two high pressures, there was long-term, large-scale, sunny and hot weather in this region [54].

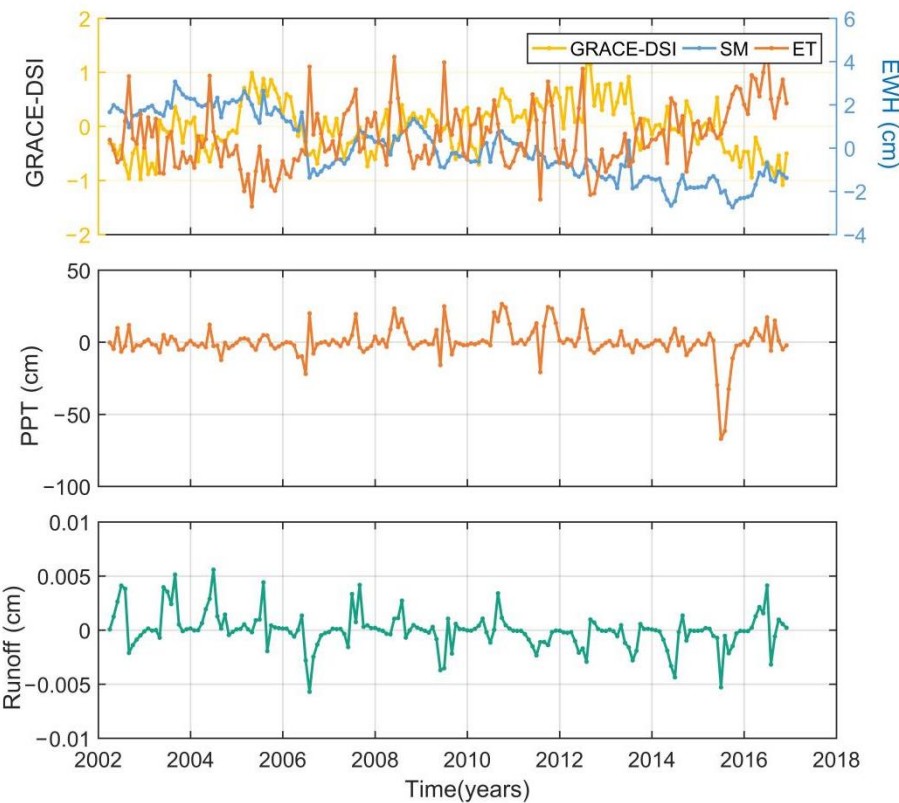

**Figure 10.** The time series of GRACE-DSI/SCPDSI, PPT anomaly, SM anomaly, runoff anomaly and ET anomaly under the plateau mountain climate.

**Table 8.** The correlation coefficients between drought indices and hydrological components.

| Hydrological Component | PPT | SM | Runoff | ET |
|:---:|:---:|:---:|:---:|:---:|
| GRACE-DSI | 0.12 | 0.29 | −0.01 | −0.65 |

**Table 9.** The correlation coefficients between different hydrological components.

| Hydrological Component | PPT | SM | Runoff | ET |
|:---:|:---:|:---:|:---:|:---:|
| PPT | 1 | 0.10 | 0.17 | 0.30 |
| SM | 0.10 | 1 | 0.43 | −0.25 |
| Runoff | 0.17 | 0.43 | 1 | −0.01 |
| ET | 0.30 | −0.25 | −0.01 | 1 |

Figure 12 shows that drought events occurred during all three negative NAO events, resulting from the southward movement of the Siberian High [55]. The opposite was true during positive NAO events. However, the drought event from April 2007 to April 2008 appeared during a positive NAO event. The reason for this drought was explained in the previous paragraph.

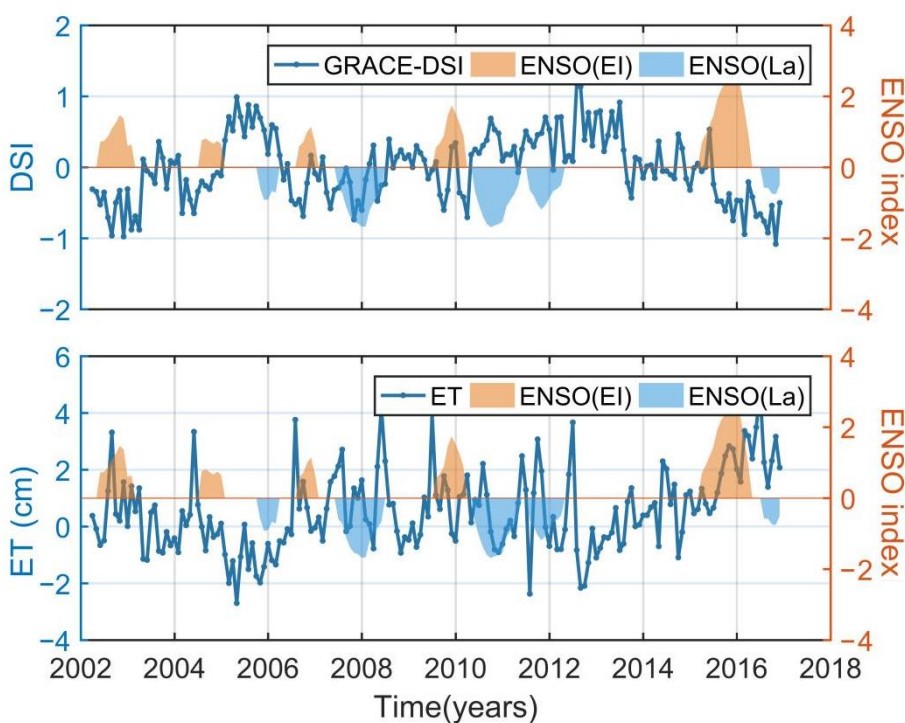

**Figure 11.** GRACE-DSI and ET anomaly compared with ENSO index (El, El Niño; La, La Niña).

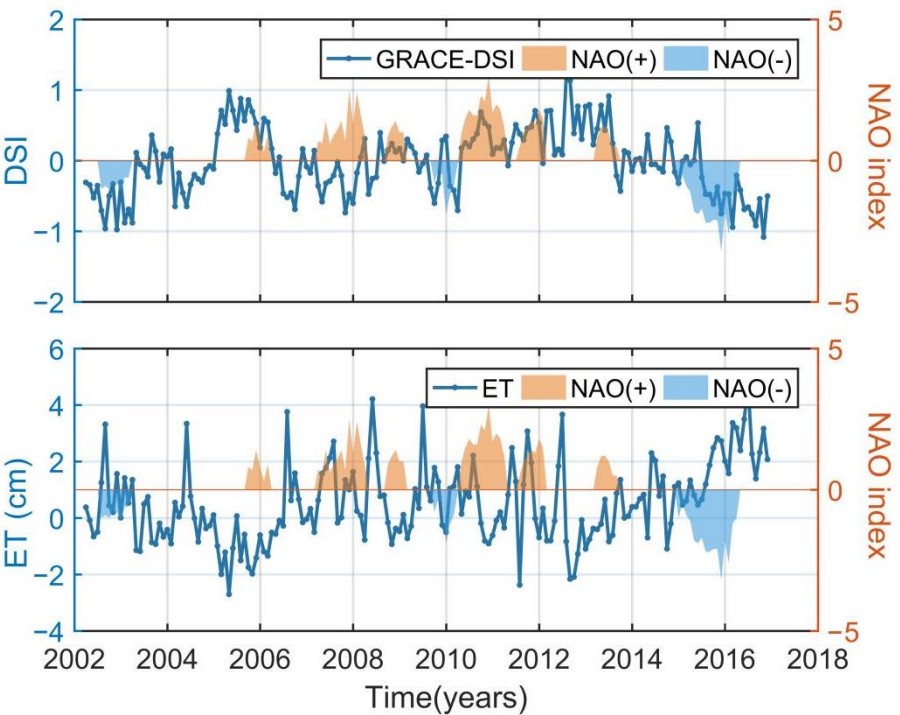

**Figure 12.** GRACE-DSI and ET anomaly compared with NAO index.

4.2.4. Temperate Continental Climate (IV)

Figure 13 compares the time series of GRACE-DSI and PPT, SM, ET and runoff anomaly under the temperate continental climate. We found that the GRACE-DSI had a significant correlation with both ET and SM. Additionally, the change trend of the PPT anomaly was relatively stable. The absolute value of the correlation coefficient between ET and GRACE-DSI (0.76) was greater than that of GRACE-DSI and PPT (0.57) (Table 10). This indicates that the impact of ET on drought events is greater than that of PPT in the

temperate continental climate in this climate. We calculated the correlation coefficients between the four hydrological components (SM, ET, PPT and runoff), as shown in Table 11. The results also show the same relationships between PPT, runoff and SM.

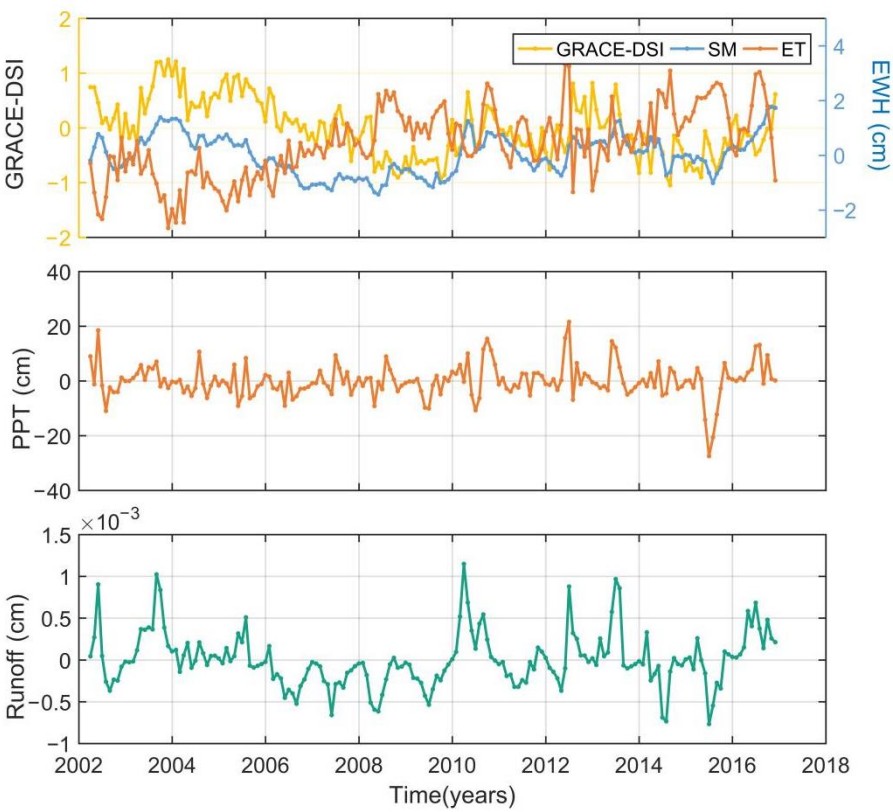

**Figure 13.** The time series of GRACE-DSI/SCPDSI, PPT anomaly, SM anomaly, runoff anomaly and ET anomaly under the temperate continental climate.

**Table 10.** The correlation coefficients between drought indices and hydrological components.

| Hydrological Component | PPT | SM | Runoff | ET |
|---|---|---|---|---|
| GRACE-DSI | 0.12 | 0.57 | 0.37 | −0.76 |

**Table 11.** The correlation coefficients between different hydrological components.

| Hydrological Component | PPT | SM | Runoff | ET |
|---|---|---|---|---|
| PPT | 1 | 0.29 | 0.53 | 0.21 |
| SM | 0.29 | 1 | 0.66 | −0.35 |
| Runoff | 0.53 | 0.66 | 1 | −0.15 |
| ET | 0.21 | −0.35 | −0.15 | 1 |

The time series of GRACE-DSI, PPT and ET anomaly during ENSO and NAO events are shown in Figures 14 and 15. Figure 14 shows that drought events occurred during three El Niño events. The previous studies show that the influence of ENSO events in the temperate continental climate region is basically the same as that in the temperate monsoon climate region [50,51]. However, there was no drought during the other two El Niño events, because the degree of anomalous decrease in PPT was small and the ET was less than usual. Drought events occurred during two La Niña events. The Subtropical High that moved eastward from Western or Central Asia under the influence of atmospheric circulation controlled this region and was the main factor causing the drought event from July 2007 to June 2008 [69]. The drought event from August 2011 to April 2012 was mainly

caused by abnormal atmospheric circulation. As the Subtropical High was weaker than usual, the region was mainly controlled by cold air, and warm and humid air currents could not reach the area. This led to reduced PPT and drought [56].

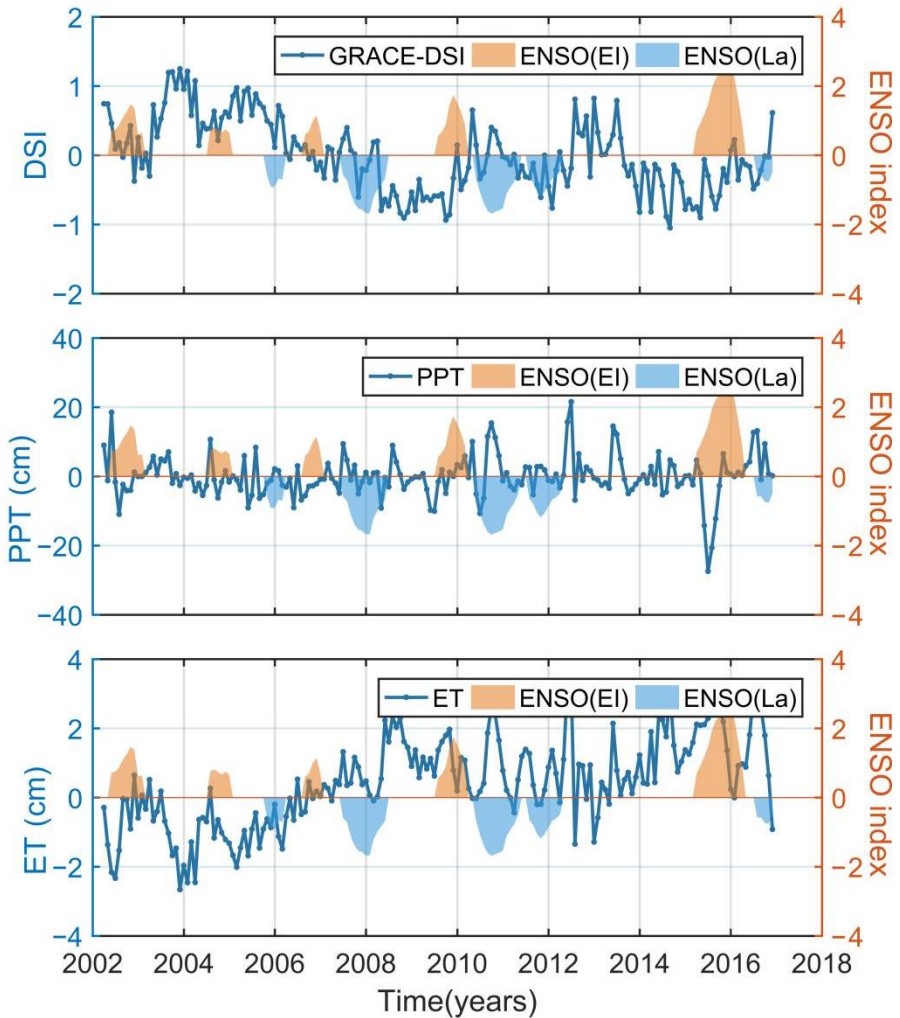

**Figure 14.** GRACE-DSI, PPT and ET anomaly compared with the ENSO index (El, El Niño; La, La Niña).

From Figure 15, two drought events appeared during negative NAO events. Although the PPT was less than usual, ET was also less; this is why there was no drought during the negative event from August 2008 to March 2003. Drought only appeared during two positive NAO events. The drought from July 2015 and April 2016 was affected by El Niño events. As the region was affected by the Subtropical High, the cold air flowed southward. At the same time, due to the influence of the sinking airflow, the transportation of water vapor in the south was blocked. This caused a drought event from October 2008 to February 2009 [70].

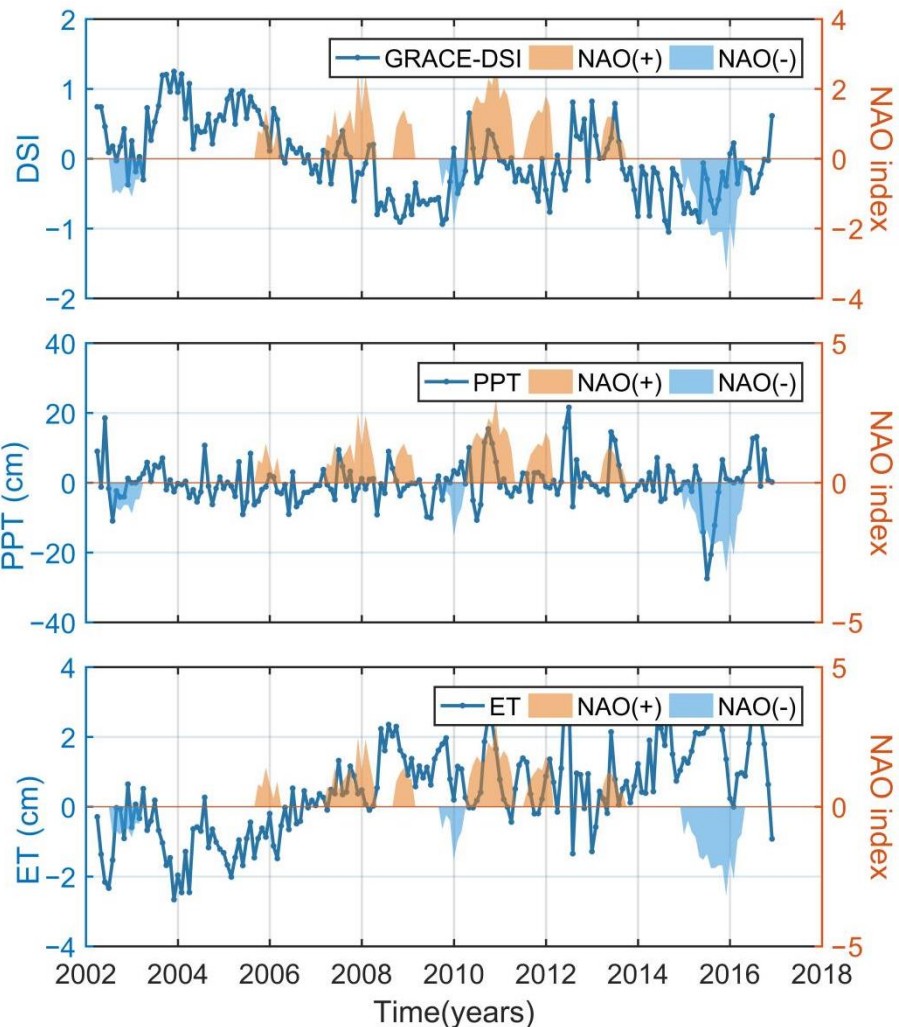

**Figure 15.** GRACE-DSI, PPT and ET anomaly compared with the NAO index.

## 5. Discussion

PPT and ET are the two decisive factors leading to drought events. Extreme climate is one of the factors causing abnormal PPT and ET. The purpose of our study is to discuss the influence of extreme climate on GRACE-DSI, PPT and ET in different climate regions. According to the results in Section 4, extreme climate led to drought by affecting PPT and temperature. Table 12 shows relationship between extreme climate and both PPT and ET anomaly in four climate types. From Table 12, it can be seen that ENSO index was negatively correlated with PPT anomaly under the temperate continental climate and temperate monsoon climate. Meanwhile, there was a positive correlation between ENSO index and PPT anomaly under the subtropical monsoon climate (see the analysis in Section 4). The previous studies show that East Asian Monsoon weakens during El Niño events, which causes the Western Pacific Subtropical High to weaken and move to the south. When this occurs, the boundary between cold air and warm and humid air in MC moves further southward than in regular years. This increases the probability of excessive PPT in South China, while it is likely to lead to reduced PPT in North China. During La Niña events, the situation is the opposite [51,61,71].

**Table 12.** The relationship between ENSO and NAO indices and GRACE-DSI, PPT and ET anomaly.

| Correlation Coefficient | I | II | III | IV |
|---|---|---|---|---|
| ENSO vs. PPT anomaly | −0.30 | 0.27 | - | −0.30 |
| ENSO vs. ET anomaly | 0.08 | - | 0.22 | 0.08 |
| NAO vs. PPT anomaly | 0.23 | −0.29 | - | 0.09 |
| NAO vs. ET anomaly | −0.11 | - | −0.25 | −0.10 |

Table 12 shows that ENSO index and ET anomaly were positively correlated in the temperate continental climate and temperate monsoon climate. Zuo indicated that the influence of ENSO events on the temperature in China is mainly concentrated in North China. The average maximum temperature was observed to be higher in the summer and autumn than normal during El Niño events, while one was lower during La Niña events [66]. Abnormal temperature changes are directly reflected in the ET. The PPT and ET are the two important hydrological components in the terrestrial water cycle. Therefore, abnormal changes in PPT and ET lead to abnormalities in TWSC, which in turn affects the GRACE-DSI. According to Table 12, the impact of extreme climate on PPT was greater than that on ET.

Due to the geographical location, the climate of the plateau mountain climate region is more strongly affected by the Indian Ocean Monsoon. Although blocked by the Himalaya, the Indian Ocean Monsoon still brings some warm and humid air to this region. In El Niño years, the surface temperature of the Qinghai-Tibet Plateau is relatively high due to the weakening of the Indian Ocean Monsoon, resulting in an increase in ET. In La Niña years, there is less ET in this region [65]. Therefore, the results in Table 12 also confirm the above conclusion. As the region with temperature continental climate is located in the hinterland of the continent and far from the ocean, it is difficult for warm humid air to reach this region, causing the region to be mostly semi-arid and arid. Therefore, the biggest difference between this region and the temperature monsoon climate is that ET plays a key role in the entire terrestrial water cycle.

## 6. Conclusions

To discuss the cause of drought events under different climatic conditions, we took four climatic regions in MC as an example to study the influences of various hydrological components and extreme climate on drought events using GRACE-DSI data. Firstly, we compared the temporal and spatial distribution of GRACE-DSI and SCPDSI and calculated the correlation coefficient between these two indices. The results indicate that GRACE-DSI and SCPDSI have similar temporal and spatial distribution and are strongly correlated (0.66) in MC. This proves that GRACE-DSI can detect drought events in MC. Secondly, we studied the influence of drought-related factors and extreme climate on drought events by analyzing the relationship between GRACE-DSI and anomalous changes of SM, PPT, ET and runoff during ENSO and NAO events. The results show that PPT and ET are the main causative factors of drought events. However, they play different roles in different climate regions. PPT plays a major role in temperate monsoon climate and subtropical monsoon climate regions, while ET plays a dominant role in the other two climate regions. ENSO and NAO events first affected the monsoon and regional high, and changes in the monsoon and regional high directly affected the changes of PPT and temperature in a given region. The changes of temperature affected the amount of ET. Therefore, extreme climate has a very important influence on regional drought events. However, the influence of extreme climate on different climatic regions is different depending on the different geographical locations and influence mechanisms.

Our research helps to further reveal the causes of drought events, and provides a reference for drought research in other similar and different climate regions. We mainly focused on the influence of climatic factors on drought events in this paper. Therefore, future work will mainly study the influence of human factors on drought events and the

formation mechanisms of drought in key regions, such as the North China Plain, Northwest China, the Ganges River basin, the Middle East, etc.

**Author Contributions:** Conceptualization, Z.L.; methodology, L.C.; software, C.Z. and L.C.; validation, L.C. and C.Y.; data curation, L.C. and Z.L.; writing—original draft preparation, L.C. and Q.L. writing—review and editing, L.C. and X.W.; funding acquisition, C.Y. All authors have read and agreed to the published version of the manuscript.

**Funding:** This research was funded by the National Key R&D Program of China (Grant No. 2018YFC1503503), the National Natural Science Foundation of China (Grant No. 41931074, 42061134007, 42004013), Foundation of Young Creative Talents in Higher Education of Guangdong Province (Grant No. 2019KQNCX009) and Open Fund of Guangxi Key Laboratory of Spatial Information and Geomatics (19-050-11-03).

**Institutional Review Board Statement:** Not applicable.

**Informed Consent Statement:** Not applicable.

**Data Availability Statement:** GRACE RL06 data: ftp://icgem.gfz-potsdam.de/01_GRACE/CSR/CSRRelease06 (accessed on 16 September 2021); GLDAS model data: https://disc.gsfc.nasa.gov/datasets?keywords=gldas&page=1 (accessed on 16 September 2021); In-situ PPT data: http://data.cma.cn (accessed on 16 September 2021); SCPDSI data: (https://crudata.uea.ac.uk/cru/data/drought/ (accessed on 16 September 2021); ENSO index data: http://www.cpc.ncep.noaa.gov/data/indices/ (accessed on 16 September 2021); NAO index data: https://www.cpc.ncep.noaa.gov/products/pre-cip/CWlink/pna/nao.shtml (accessed on 14 September 2021).

**Acknowledgments:** We are grateful to the Center of Space Research (CSR) for providing the monthly GRACE gravity field solutions, and to the Goddard Space Flight Center for providing the monthly GLDAS-2.1 data, to the China National Meteorological Science Data Center for providing the monthly precipitation and temperature products, to the Climatic Research Unit at university of East Anglia for providing the SCPDSI data and to the National Oceanic and Atmospheric Administration for providing the Extreme weather Index Data.

**Conflicts of Interest:** The authors declare no conflict of interest.

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
