# Peer review of "Analysis of the Influencing Factors of Drought Events Based on GRACE Data under Different Climatic Conditions: A Case Study in Mainland China"

_water, doi:10.3390/w13182575_

Round 1

Reviewer 1 Report

Comments on Analysis of the Influencing Factors of Drought Events Based on GRACE Data Under Different Climatic Conditions: A Case Study in Mainland China:

  • Lines 41-43, cite also these recent useful papers to improve the literature and to show the importance of your work:

Spatial and Temporal Analysis of Rainfall and Drought Condition in Southwest Xinjiang in Northwest China, Using Various Climate Indices

Global surface temperature: A new insight

  • Discuss the main reasons for the variations of the spatial distribution of monthly GRACE-DSI and SCPDSI concerning the October 2009-September 2010 period.
  • How can expand the results in other regions with similar/different climates?
  • At the end of the manuscript, explain the implications and future works considering the outputs of the current study.
  • The quality of the language needs to improve by a native English speaker for grammatically style and word use.

Reviewer 2 Report

Analysis of the Influencing Factors of Drought Events Based on 2 GRACE Data Under Different Climatic Conditions: A Case 3 Study in Mainland China

1- introduction: Explain the importance of studying drought in the region.

2-Figure 1: Please add a legend and smaller map to Figure 1 and show the geographical location of the study area.

3-Results and Analysis: There is no uniform design in the graphs and they look irregular. Please spend enough time and attention on them.

4- It is better to do a comparative evaluation for other sections like 4.1. Comparison of GRACE-DSI and SCPDSI reduces the confusion in the content because the reader will be confused.

5- You have done a lot of work and this is valuable, but you should summarize your work in the best way so that the reader is not confused. Re-categorize the sections.

Author Response

The response to Reviewer is in the attachment.

This manuscript is a resubmission of an earlier submission. The following is a list of the peer review reports and author responses from that submission.